# Effect of Fly Ash Belite Cement on Hydration Performance of Portland Cement

Yongfan Gong [1,2,*], Jianming Yang [1], Haifeng Sun [1] and Fei Xu [3]

[1] College of Civil Science and Engineering, Yangzhou University, Yangzhou 225127, China; MX120200575@yzu.edu.cn (J.Y.); MZ120190754@yzu.edu.cn (H.S.)
[2] Jiangsu Huatai Road and Bridge Construction Group Co., Ltd., Yangzhou 225000, China
[3] College of Civil Engineering, Yangzhou Vocational University, Yangzhou 225000, China; 101831@yzpc.edu.cn
[*] Correspondence: yfgong@yzu.edu.cn; Tel.: +86-1585-066-3317

**Abstract:** Fly ash belite cement is a green, low carbon cementitious material, mainly composed of hydraulic minerals of dicalcium silicate and calcium aluminate. In this study, we used fly ash belite cement to control the setting time, hydration heat, strength, composition and microstructure of hydration products in Portland cement. Results showed that incorporating fly ash belite cement into Portland cement can shorten the setting time, accelerate hydration reaction speed, enhance early hydration heat release rate of silicate minerals and reduce total hydration heat. Moreover, replacing composite cement with 30% FABC causes the 90 d compressive strength of pastes and mortars to reach 107 and 46.2 MPa, respectively. The mechanical properties can meet the requirements of P·F 42.5 cement. During the hydration reaction process, clinker and Portland cement have a synergistic hydration effect. Notably, hydration of fly ash belite cement promotes the formation of C-S-H gel, Ettringite and calcium hydroxide, thereby significantly enhancing long-term strength. With the increase of FABC contents, the long-term strength would be improved with the densification of hydration products. The porosity has a great influence on the strength, and the high porosity was the main cause of the low early strength of FABC pastes. FABC and its composite cement show promise for mass concrete applications and can be applied as a setting agent for Portland cement.

**Keywords:** belite; fly ash; hydration heat; mechanical properties

## 1. Introduction

The annual output of fly ash across key industrial enterprises in China was about 540 million tons, with a comprehensive utilization of 410 million tons, in 2019. However, the utilization of storage capacity in previous years was only 2.13 million, while the comprehensive utilization rate was 75.9%. Based on these statistics, it is evident that not only has the stockpiled low-quality fly ash been underutilized over the years, but its accumulation is still growing. Large amounts of fly ash not only occupy numerous land resources, but also produce dust that pollutes the atmosphere. Discharge into the water system causes silting of rivers, while toxic chemicals present in the fly ash, such as cadmium, mercury, lead, chromium and arsenic elements, among others, can also be harmful to human and plant life [1]. Fly ash belite cement (FABC), which is prepared from fly ash and lime by hydrothermal synthesis and low-temperature calcination (<900 °C), contains hydraulic minerals of calcium silicate ($2CaO \cdot SiO_2$; $\beta$-$C_2S$) and calcium aluminate ($12CaO \cdot 7Al_2O_3$; $C_{12}A_7$). Previous studies have described it as a new type of green low carbon cementitious material [2–5]. Its mineral components, $\beta$-$C_2S$ and $C_{12}A_7$, which are formed at low temperature, are in a metastable state, and possess hydration rates, setting times, strength and other properties that are significantly different from those of Portland cement [6,7].

$Na^+$, $K^+$ and $P^{5+}$ can stabilize the $\beta$-$C_2S$ structure, reduce viscosity and surface tension of clinker during sintering, thereby improving the quality of clinker $\beta$-$C_2S$ activity.

Shahsavari found that the early hydration rate of $C_2S$ mainly depends on the number and nature of defects [8]. For example, β-$C_2S$ synthesized at low temperature has higher activity than that synthesized at high temperature, indicating that changes in temperature and ion concentration can improve the activity and early strength of calcium silicate minerals. Moreover, Sanchez et al. [9] evaluated the effect of hydration medium on hydration behavior of $C_3S$ and $C_2S$, and found that the addition of NaOH solution accelerated hydration of both minerals, and just improved the early strength of $C_2S$ hardened paste. Notably, the 28-d compressive strength of hardened paste increased from 8.8 to 18.2 MPa, although it adversely affected the ultimate strength of $C_3S$. On the other hand, Thomas et al. [10] analyzed the effects of sodium silicate and C-S-H gels on hydration rate and activation energy of $C_2S$, and found that the addition of activators increased the activation energy of $C_2S$ from 32 to 55 kJ/mol. The authors concluded that the observed change from low to high activation energy affirmed that sodium silicate can accelerate the dissolution of calcium silicate as well as nucleation and growth rates of hydration products. Previous evidence indicates that the hydration activity of $C_2S$ is low under ordinary hydration conditions. In fact, $C_2S$ synthesized at low temperature has been found to accelerate the reaction rate of silicate and aluminum phase minerals in Portland cement, thereby significantly improving the early hydration performance of Portland cement.

Several researchers [11–14] have adopted mechanical and chemical activation, in the hydrothermal synthesis method, to achieve the common excitation of surface and chemical energy of low-quality fly ash vitreous particles. This "superposition effect" can efficiently promote hydrothermal synthesis reaction of active components and lime, accelerate the hydrothermal synthesis rate, and improve the specific surface area, reactivity as well as the total content of cementitious minerals. To date, most research works [15–17] have focused on improving the hydrothermal synthesis process, while little is known regarding the mineral structure, properties and synergistic hydration mechanism. In this paper, we analyze the synergistic hydration and performance of FABC and Portland cement, and reveal the activity and function of $C_2S$ and $C_{12}A_7$ synthesized at low temperatures. Our findings provide new insights to guide the utilization of low-quality fly ash.

## 2. Materials and Methods

### 2.1. Raw Materials

Fly ash belite cement were locally made, in our laboratory, whereas Portland cement of Chinese P·I 42.5 cement was produced by Yangzhou Lvyang Cement Co., Ltd., ISO679-standard sands (Xiamen ISO Standard Sand Co.,Ltd, Xiamen, China), before being used to test the strength of cement mortars. Gypsum (Analytical Reagent, $CaSO_4 \cdot 2H_2O \geq 99.0\%$) was acquired from Chengdu Chron Chemicals, and its design proportions were added to FABC. The main active components in clinker are β-$C_2S$ and $C_{12}A_7$, although it also contains a small amount of mullite and low-temperature quartz as well as other inactive minerals. The total amount of hydraulic minerals in the clinker was 54.5%, after quantitative analysis using X-ray diffraction phase (K value method). A summary of the chemical composition of fly ash belite and Portland cement is shown in Table 1, whereas mixing proportions for the composite cement are shown in Table 2.

**Table 1.** Chemical composition of FABC and P·I 42.5.

| No. | Chemical Composition /wt-% | | | | | | | | | |
|---|---|---|---|---|---|---|---|---|---|---|
| | SiO$_2$ | Al$_2$O$_3$ | Na$_2$O | Fe$_2$O$_3$ | CaO | TiO$_2$ | K$_2$O | MgO | SO$_3$ | Loss |
| FABC | 32.94 | 19.68 | 1.06 | 3.80 | 36.87 | 0.83 | 0.71 | 0.82 | 0.53 | 1.97 |
| P·I 42.5 | 22.04 | 4.76 | 0.53 | 3.10 | 64.5 | N/A | 0.32 | 0.92 | 1.90 | 1.01 |

**Table 2.** Mix proportions of composite cement.

| Sample | FABC/% | P·I 42.5 |
|--------|--------|----------|
| RC | 0 | 100 |
| FAC10 | 10 | 90 |
| FAC20 | 20 | 80 |
| FAC30 | 30 | 70 |
| FAC40 | 40 | 60 |
| FAC50 | 50 | 50 |

*2.2. Methods*

Composite cement pastes, with different contents of FABC (w/c = 0.30) were prepared and used to generate cubic specimens measuring $20 \times 20 \times 80$ mm$^3$. On the other hand, composite cement mortars with different contents of FABC were prepared with an W/C (water to cement) of 0.5, to generate specimens measuring $40 \times 40 \times 160$ mm$^3$, with an S/C (sand to cement) of 3:1. The specimens were cured after 24 h, using water at temperature $20 \pm 1$ °C and relative humidity of 90%. Thereafter, their properties were tested after 3, 28, and 90 days from GB/T17671-1999 (ISO method). Standard consistency water demand and setting times of the composite cement were tested from GB/T1346-2011.

Hydration products of hardened pastes were cured for 3, 28, and 90 days, then analyzed using an Isothermal Calorimetry (TAM-Air 8, TA Instruments, New Castle, DE, USA), X-ray diffraction (XRD, Smart Lab, Rigaku, Tokyo, Japan) with CuKα radiation (30 kV and 20 mA) at a scanning rate of 4°/min, Fourier transform infrared spectrometer (FTIR, Nicolet 6700, Thermo Scientific, Waltham, MA, USA) by the KBr method with a resolution of 0.1 cm$^{-1}$, and Thermogravimetric analysis (TG, Pyris 1 TGA, Waltham, MA, USA) in a dynamic nitrogen stream (flow rate = 100 cm$^3$/min) at a heating rate of 10 °C/min, to determine the chemical composition of minerals and hydration products across different times. Morphology and pore structure of the products were characterized using Field emission scanning electron microscopy images (SEM, Hitachis-4800, Tokyo, Japan) at a working voltage of 15 kV.

**3. Results**

*3.1. Effect of FABC on Hydration Rate of Portland Cement*

3.1.1. Setting Time

The setting time of Portland cement with different FABC contents is shown in Figure 1. An increase in FABC contents, from 10 to 50%, caused a significant decrease in initial and final setting times of Portland cement. Specifically, the initial setting time decreased from 205 to 45 min, while the final setting time decreased from 315 to 105 min, representing 78 and 66% reduction, respectively. This was mainly attributed to higher activity and content of $C_{12}A_7$, as well as a faster reaction speed of FABC, in FABC than in $C_3A$ in ordinary Portland cement (OPC). At the same time, the calcination of hydrated C-S-H gels and calcium aluminate calcined at low temperature resulted in smaller particle size and larger surface area, relative to clinkers obtained at a higher temperature. Notably, these low-temperature clinkers immediately reacted with water, causing a "dry" effect on the slurry [18], which further simultaneously accelerated the setting and hardening.

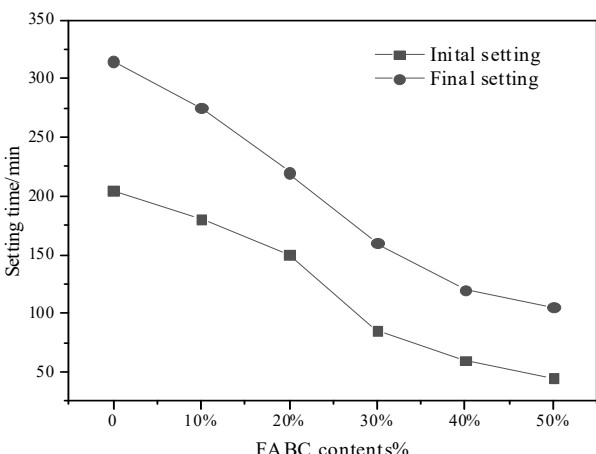

**Figure 1.** Setting time of cement with different FABC content.

3.1.2. Hydration Heat

The total hydration heat release of Portland cement across different FABC contents is shown in Figure 2. In summary, an increase in FABC caused a significant increase and decrease in early and total heat release of hydration, respectively. At 3 days, the total hydration heat of Portland cement reached 239 J/g, which was three times higher than that of FABC. FABC's hydration heat at 12 h was 82.5% that recorded at 3 d, whereas hydration heat of Portland cement at 12 h just reached 26.8% of the total hydration heat of 3 d. This was attributed to the faster hydration rate of $C_{12}A_7$ in FABC compared to that of Portland cement. Moreover, a large amount of hydration heat was released over a short period, the setting time of $C_3S$ and $C_2S$ was longer in Portland cement, which allowed a large amount of hydration heat to be gradually released after 12 h. The addition of 20 and 30% FABC caused the total heat of hydration of Portland cement to increase to 81 and 88 J/g, respectively, after 12 h, representing a 39.3 and 42.2% increase in the total heat of hydration at 3 d. This was also significantly higher than the total hydration heat of Portland cement at the same time point. These results indicate that FABC has a process of cooperative hydration with Portland cement, and changes the hydration rate of composite cement. This phenomenon may be due to the formation of $Ca(OH)_2$ from the hydration of silicate clinker. Consumption of large amounts during the hydration process of $C_{12}A_7$ breaks the solid–liquid balance of $C_3S$ and $C_2S$ hydration, thereby improving early hydration of $C_3S$ and $C_2S$, and increasing the early hydration exothermic rate of composite cement [19].

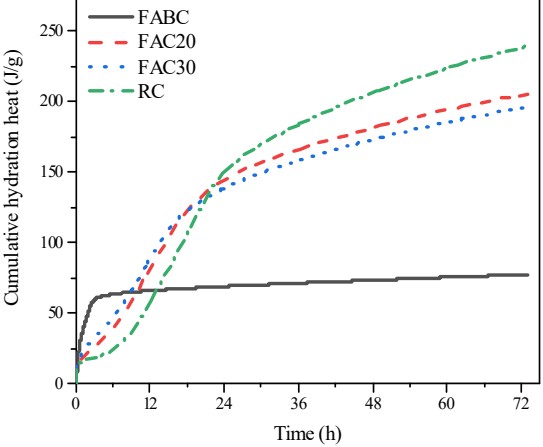

**Figure 2.** Hydration heat of cement with different FABC content.

### 3.2. Effect of FABC on the Strength of Portland Cement

3.2.1. Strength of Pastes

Flexural and compressive strengths of Portland cement pastes, with different contents of FABC, are illustrated in Figure 3. In summary, FAC30 paste has the highest 90-day flexural compressive strength, which reached 11.8 and 107 MPa, respectively. Increasing FABC content from 10 to 50% caused a reduction in flexural and compressive strengths of 3-days pastes. An FABC content of less than 40% caused the flexural and compressive strength guarantee rates of the paste samples to reach 98.8 and 90.4%, respectively, after 28 d. Notably, at 90 d, the compressive strength guarantee rate of FAC30 reached 110%. This was mainly because during the early stages of hydration, the proportion of $C_3S$ in FABC decreased, hence it was unable to provide enough early strength hydraulic minerals. At the same time, the Ettringite generated by $C_{12}A_7$ hydration produced micro-expansion and micro-cracks which caused a decrease in strength. Conversely, the proportion of $C_2S$ hydration products in FABC pastes increased during later hydration stages. Silicate minerals in Portland cement are stimulated by $C_2S$-$C_{12}A_7$ clinker minerals. Notably, more $Ca(OH)_2$ contents were generated after hydration, improving the density of the structure, the development of strength was better in the later stages than of Portland cement.

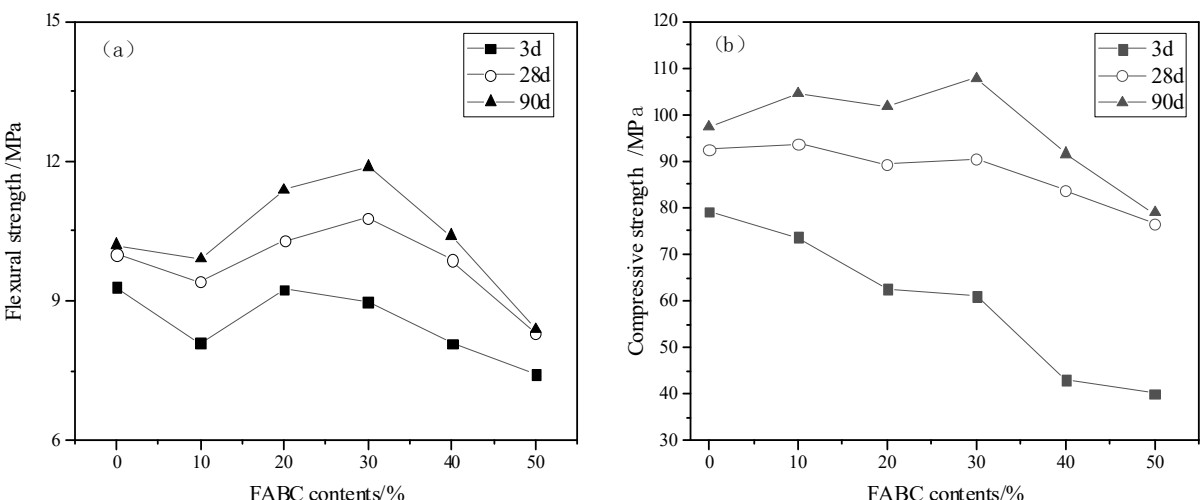

**Figure 3.** The strength of pastes with different FABC content: (**a**) Flexural strength; (**b**) Compressive strength.

3.2.2. Strength of Mortars

Flexural and compressive strengths of Portland cement mortars with different contents of FABC are illustrated in Figure 4. From the results, it is evident that increasing FABC contents to 40% generated a slight decrease in both flexural and compressive strengths of mortars at 3, 28 and 90 d. When the FABC content was less than 20%, the guarantee rates of both flexural and compressive strengths of mortars could reach more than 90% at 3 d. However, an FABC content of more than 20% resulted in a marked decrease in early flexural and compressive strengths of mortars, as well as the rapid development of later strength. Moreover, compressive strengths for FAC30 and FAC40 reached 46.2 and 44.6 MPa respectively, at 90 d, these were represented 93.3 and 90.1% of RC at the same period, respectively. This was mainly attributed to the occurrence of more pores on the surface of clinker minerals, as well as a 20% increase in the volume porosity of early hydration hardening paste [20]. Moreover, a decrease in internal pores of the specimen, coupled with no dense formation of the structure, caused a reduction in the early strength. However, at 90 d, the specimen's internal pores decreased, causing a gradual improvement in the strength of mortar specimens during late hydration. It was indicated that [17] the high porosity was the main cause of the low early strength of FABC pastes, and it was also not conducive to the development of the early strength of composite cement. However,

with the increase of FABC contents, the long-term strength would be improved with the densification of hydration products. In conclusion, the long-term strength of motars will obviouly increase when the proportion of FABC is less than 30%.

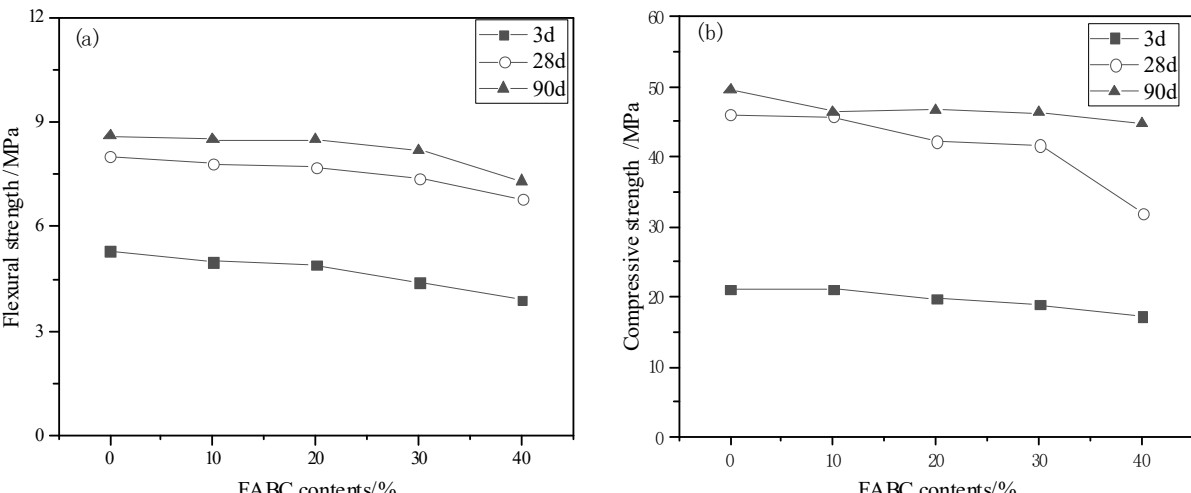

**Figure 4.** The strength of mortars with different FABC content: (**a**) Flexural strength; (**b**) Compressive strength.

### 3.3. Effect of FABC on Hydration Products of Portland Cement

#### 3.3.1. FTIR Spectrum and X-ray Diffraction Spectrum

The FTIR spectrum of composite cement hardened paste at 28 d is shown in Figure 5a. From the spectrum, the absorption band at 3430 $cm^{-1}$ is the asymmetric stretching vibration of $OH^-$ (v3), which indicates the presence of Ca $(OH)_2$ in the hardened paste. Increasing FABC content increased the asymmetric stretching vibration of $CO_3^{2-}$, to 1419 $cm^{-1}$ (v3), indicating marked carbonization in the pastes. Asymmetric stretching vibrations of Si-O-Si (v3), at 956 and 873 $cm^{-1}$, indicated the presence of more C-S-H gels with disordered structure and multiple polymerization states [21]. Notably, increasing FABC content caused a marked absorption band of Si-O-Si, indicating a gradual increase in the polymerization degree of Si-O bonds as well as hydration products. This phenomenon may be attributed to because a direct release of Ca $(OH)_2$ from $C_{12}A_7$, which rapidly reacted with $C_{12}A_7$ to form Ettringite at low calcium content in pastes. In order to maintain the solid-liquid equilibrium, there is a need to ensure that hydration of Portland cement is in the affirmative direction, and C-S-H hydration products are gradually increased. The reaction equation is shown in Formulas (1) and (2). Figure 5b shows the XRD spectrum of composite cement hardened paste at 28 d. Although C-S-H gel and Ettringite ($Ca_6Al_2(OH)_{12}(SO_4)_3 \bullet 26H_2O$, AFt, $2\theta = 9.2°$) are generated in large quantities, there are still un-hydrated silicate clinker minerals in the hardened paste, while mullite and quartz particles from fly ash ($2\theta = 25 \sim 28°$) are abundant [22]. Based on this, it is evident that the characteristic peak of Ca $(OH)_2$ in the XRD spectrum first increases, then decreases with an increase in FABC content. This result further corroborates the findings on the strength of the pastes. The reaction of $Ca^{2+}$, $SO_4^{2-}$, Al $(OH)_4^-$ generating to AFt would consume a lot of Ca $(OH)_2$, which would destroy the solid–liquid equilibrium of $C_3S$ and $C_2S$. It promotes the hydration of pastes, releases more $Ca^{2+}$, and finally improves the content of hydration products. Hence, the forming of Ettringite is beneficial to the hydration of composite cement.

$$2CaO \cdot SiO_2 + nH_2O \rightarrow xCaO \cdot SiO_2 \cdot yH_2O + (2 - x)Ca(OH)_2 \tag{1}$$

$$12CaO \cdot 7Al_2O_3 + 21CaSO_4 \cdot 2H_2O + 9Ca(OH)_2 \rightarrow 7(3CaO \cdot Al_2O_3 \cdot 3CaSO_4 \cdot 32H_2O) \tag{2}$$

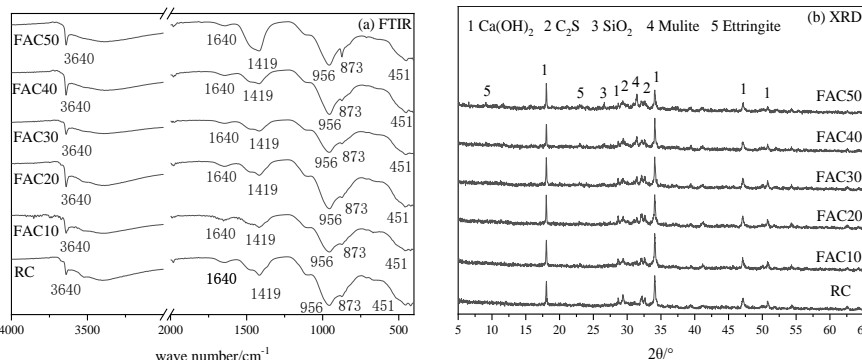

**Figure 5.** FTIR and XRD spectrum of pastes at 28 d: (**a**) FTIR; (**b**) XRD.

### 3.3.2. Thermal Analysis

Results from thermal analysis of composite cement hardened pastes at 3 d are shown in Figure 6. In summary, the first weight loss occurs at 80~200 °C, and this is mainly caused by the decomposition of C-S-H gel, Ettringite and other hydration products. Mass losses of RC, FAC10, FAC20 and FAC30 hardened pastes are found to be 6.61, 6.76, 7.59 and 8.51%, respectively, indicating that hydration of products increases with an increase in FABC content. The second weight-loss stage occurs at 400–500 °C, and mainly comprises the decomposition of Ca $(OH)_2$. Notably, water loss ratios for Ca $(OH)_2$ in RC, FAC10, FAC20 and FAC30 hardened pastes were found to be 2.88, 3.07, 3.19 and 2.55%, respectively. Although an increase in FABC content enhanced the formation of Ca $(OH)_2$, content above 30% caused a marked decrease in total Ca $(OH)_2$. This is attributed to the fact that hydrated products of $C_{12}A_7$ react with gypsum and calcium hydroxide to form Ettringite. The third weight-loss stage occurs at 600–800 °C, and mainly involves the decomposition of $CaCO_3$ due to carbonization. The mass-loss rate of CaCO3, after decomposition, is FAC30 > FAC10 > RC > FAC20. In summary, FABC increased the total amount of hydration products, such as C-S-H gels, Ettringite and calcium hydroxide, and accelerated the early hydration rate of Portland cement.

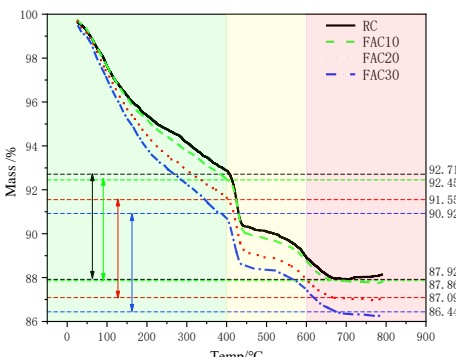

**Figure 6.** Thermal analysis of pastes at 3 d.

### *3.4. Effect of FABC on Micro-Structure of Portland Cement Pastes*

The micro morphology of hardened pastes of Portland cement and composite cement FAC20, at 28 d, is summarized in Figure 7. In summary, hydration products of Portland cement exhibit a relatively close network structure after 28 d of hydration. As Diamond proposed [23], the morphology type of phases in cement has to be established. $C_3S$ and $C_2S$ were almost hardly visible on the surface of pastes, and C-S-H gels were showed as a network structure which is belongs to type II hyration products. Moreover, there are numerous pores on the surface of C-S-H gel in the hydration products of FAC20, while the calcium hydroxide and Ettringite crystals show a marked increase in size. These Ettringite crystals occur in the pores of hydrated calcium silicate gel in disorder, filling the pore

structure and increasing the density of hardened paste. This enhances the development of later strength in the pastes. The main reason for this phenomenon is that the addition of FABC into Portland cement causes $Ca(OH)_2$ to react with $Al(OH)_4^-$ released from $C_{12}A_7$ to form $C_3AH_6$ and Ettringite, thereby accelerating hydration of $C_3S$ and breaking the solid–liquid equilibrium of $C_3S$, $C_2S$ and $Ca(OH)_2$ solution as well as silicate minerals, and shortening the setting time. However, $C_{12}A_7$ is a surface porous spherical structure, with a high specific surface area and high activity. Hardened paste has a higher porosity and a lower early strength, while elongating the time period causes a gradual increase in the late strength. Therefore, the composite cement with fly ash belite and Portland cement are effective components that can play a synergistic hydration role to accelerate the hydration of cement and the formation of hardened paste structure.

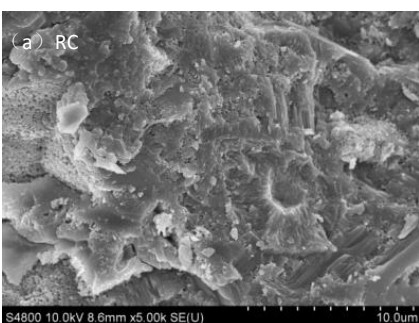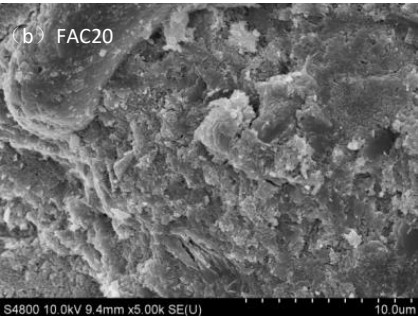

**Figure 7.** Micro-structure of Portland cement and composite cement pastes at 28d age: (**a**) RC; (**b**) FAC20.

## 4. Discussion

Results of this study indicated that $C_{12}A_7$ has a marked effect on the evolution of cementitious material paste across the rheological, and plastic to hardening states. In particular, it can significantly accelerate the reaction rate of both $C_2S$ and $C_3S$ in composite cement with the increase in late strength and shortened setting time attributed to a synergistic effect by two factors.

### 4.1. Hydration Medium Improves the Hydration Performance of $C_2S$

$C_2S$'s early hydration rate mainly depends on the number and properties of defects, while dislocation nuclear energy is more important than dislocation density [24]. Moreover, high diffusivity, fluidity and surface energy can be obtained by doping activator in the $C_2S$ crystal structure, and cause a significant improvement in hydration activity. In the present study, FABC was prepared by adding some Na+ as an alkali activator. Notably, active $C_2S$ accelerated the rate of reaction of both silicate and aluminum phase minerals in Portland cement, which significantly improved its early hydration performance. Despite the low hydration activity of $C_2S$, under ordinary hydration conditions, we noted a significant improvement under alkaline excitation, as well as faster $C_2S$ hydration rate, and significantly improved early strength. Then, the C-S-H gels can be used to speed up the reaction rate as a crystal seed, which can induce the hydration of $C_3S$ in Portland cement.

### 4.2. Promoting the Effect of $C_{12}A_7$ on Hydration and Hardening of Silicate Minerals

The mechanism of the amorphous $C_{12}A_7/CaSO_4\cdot2H_2O$ system has a powerful influence on the performance of ordinary Portland cement [25]. Specifically, the addition of 5% of the $C_{12}A_7/CaSO_4\cdot2H_2O$ system (the mass ratio of $C_{12}A_7$ to gypsum was 1:1) caused the initial and final setting times to decrease by 33 and 36% respectively, relative to a blank sample, whereas the 7-d compressive strength of the paste specimen increased from 58 to 100 MPa. Therefore, it is evident that a combination of $C_{12}A_7$ and gypsum can significantly accelerate the hydration of Portland cement, and generate a dense structure of cement paste. Moreover, a system composed of $C_{12}A_7$ and gypsum can promote the hydration of

silicate minerals and accelerate the reaction rate of $C_2S$ in composite cementitious materials. However, further research is needed to elucidate the coexistence ratio of $C_{12}A_7$. This will improve hydration efficiency and rate of $C_2S$, as well as the products after hydration and early mechanics properties.

## 5. Conclusions

The analysis of hydration properties of composite cement, predominantly composed of OPC and FABC, revealed significant differences in setting times, hydration heat, strength, as well as composition and micro-structure with Portland cement. From our experimental results, the following conclusions and recommendations are drawn:

(1) FABC can be used as a setting agent for Portland cement. The addition of 50% fly ash belite cement shortens the initial and final setting times of Portland cement to 45 and 105 min, respectively, which still meets its setting time requirements in GB/1346-2011.

(2) FABC reduces early, but enhances later, the strength of Portland cement. At 90 d, flexural and compressive strength of Portland cement paste with 30% FABC is 11.8 MPa and 107 MPa respectively; at 90 d, the compressive strength of Portland cement mortars with 30% FABC is 46. 2 MPa. These mechanical properties meet the requirements of P·F 42. 5 cement.

(3) FABC promotes hydration of Portland cement, and early hydration heat release rate of silicate minerals. However, it reduces levels of total hydration heat release in composite cement. Therefore, FABC and its composite cement show promise for mass concrete applications.

**Author Contributions:** Y.G., J.Y., H.S. are the main contributors to this research work. They carried out the main experimental program, analyzed the experimental results and drafted the research paper; F.X. mainly processed experimental data and proofread the paper. All authors have read and agreed to the published version of the manuscript.

**Funding:** This work was supported by the policy guidance program (Cooperation of Industry, Education and Academy) of Jiangsu Province (BY2020661), the key planning projects (Social Development) in Guangling District of Yangzhou City (GL202027) and the projects of Yangzhou modernizing construction industry supported by special guiding funds (201911).

**Data Availability Statement:** The data could be obtained from the corresponding author.

**Acknowledgments:** Not applicable.

**Conflicts of Interest:** The authors declare no conflict of interest.

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
