# Peer review of "Effect of Fly Ash Belite Cement on Hydration Performance of Portland Cement"

_crystals, doi:10.3390/cryst11070740_

Round 1

Reviewer 1 Report

The paper deals with the application of the fly ash as curing additive for the Portland cement using several techniques (XRD, SEM, TG, FT-IR). Some part of the work is maybe of interest, especially the green goal of the work, the utilization of this industrial waste in a sustainable form, but, the manuscript in its present form requires major revision. The corrected version of the manuscript can only be accepted, if all remarks are answered carefully.

  1. The reviewer ask the authors to avoid the use of the abbreviation without accurate description especially in the abstract section.
  2. From the experimental section, there are several missing important technical data; the wavelength of the X-ray source and scanning speed of the X-ray diffractometry measurements, heating rate and atmosphere of the termogravimetric measurements, the applied resolution and accessory of the infrared analysis.
  3. For the interpretation of the infrared spectra the reviewer would like to see the used references.
  4. On the X-ray diffraction pattern of the FC40 and 50 solid (more correctly as FCA40 and 50) on Figure 5., there are several reflections remained unidentified. I do understand that the recognition of signs at higher 2 theta angles is extremely hard, but under 25° 2 theta several reflections can be well identified. The reviewer assumed that the AFt abbreviation connected to the term alumina, ferric oxide, tri-sulfate, however it is not established in the text! The formation of the AFm (alumina, ferric oxide, mono-sulfate) phases with hydroxide or carbonate interlayer anions are also extremely likely in this environment. The reflections at around 10° − 12° and 20° − 22° − 24° 2 theta indicate the formation of the hemi- and monocarbonated forms of the hydrocalumites with aluminum and iron content as well. doi:10.1016/j.ultsonch.2016.01.026, 10.1016/j.ultsonch.2016.03.008. Please consider this and identify the corresponding reflections.
  5. The denotations for the mixture of the Portland cement and fly ash are confusing, however the FAC10-50 is used in the manuscript, on the figures the FA20, 30 (Figure 2.) and FC50-10 (Figure 5.) applied!
  6. A careful proofread of the English, which is recommended, would improve the text, especially regarding to the use of the superscript and subscript (mm3, Ca(OH)2, Na+, P5+!). However, the reviewer suspects only some error in the transformation process of the submission.

Author Response

Point 1: The reviewer ask the authors to avoid the use of the abbreviation without accurate description especially in the abstract section.

Response 1: Thank you for reminding us, we have revised it as “Fly ash belite cement is a green, low carbon cementitious material, mainly composed of hydraulic minerals of dicalcium silicate and calcium aluminate.” in line 13.

Point 2: From the experimental section, there are several missing important technical data; the wavelength of the X-ray source and scanning speed of the X-ray diffractometry measurements, heating rate and atmosphere of the termogravimetric measurements, the applied resolution and accessory of the infrared analysis.

Response 2:  Thanks for the suggestion, the reviewer is correct,  we have revised it as “ Hydration products of hardened pastes were cured for 3, 28, and 90 days, then analyzed using an Isothermal Calorimetry (TAM-Air 8, TA Instruments, New Castle, De., USA) X-ray diffraction (XRD, Smart Lab, Rigaku, Tokyo, Japan) with a CuKα radiation (30 kV and 20 mA) at a scanning rate of 4 °/min, Fourier transform infrared spectrometer (FTIR, Nicolet 6700, Thermo Scientific, Waltham, MA., USA) by the KBr method with a resoulution of 0.1 cm-1, and Thermogravimetric analysis (TG, Pyris 1 TGA, USA) in a dynamic nitrogen stream (flow rate =100 cm3/min) at a heating rate of 10 ℃/min, to determine the chemical composition of minerals and hydration products across different times. Morphology and pore structure of the products were characterized using Field emission scanning electron microscopy images (SEM, Hitachis-4800, Japan) at a working voltage of 15 kV.” in line 110-118.

Point 3:   For the interpretation of the infrared spectra the reviewer would like to see the used references.

Response 3: Thanks for the suggestion, we have cited references[21] “Yousuf, M.; Felix, L.; David, L. An X-ray diffraction (XRD) and Fourier transform infrared spectroscopic (FT-IR) characterization of the speciation of arsenic (V) in Portland cement type-V. Sci Total Environ. 1998, 224, 57-68.” the references [21] verified the analysis in line 364-365.

Point 4: On the X-ray diffraction pattern of the FC40 and 50 solid (more correctly as FCA40 and 50) on Figure 5., there are several reflections remained unidentified. I do understand that the recognition of signs at higher 2 theta angles is extremely hard, but under 25° 2 theta several reflections can be well identified. The reviewer assumed that the AFt abbreviation connected to the term alumina, ferric oxide, tri-sulfate, however it is not established in the text! The formation of the AFm (alumina, ferric oxide, mono-sulfate) phases with hydroxide or carbonate interlayer anions are also extremely likely in this environment. The reflections at around 10° − 12° and 20° − 22° − 24° 2 theta indicate the formation of the hemi- and monocarbonated forms of the hydrocalumites with aluminum and iron content as well. doi:10.1016/j.ultsonch.2016.01.026, 10.1016/j.ultsonch.2016.03.008. Please consider this and identify the corresponding reflections.

Response 4: Thanks for the suggestion, We have revised it as “Ettringite (Ca6Al2(OH)12(SO4)3·26H2O, AFt, 2θ =9.2°)”, the most phase composition have  been identified on Figure 5 in line 214.

Point 5:  The denotations for the mixture of the Portland cement and fly ash are confusing, however the FAC10-50 is used in the manuscript, on the figures the FA20, 30 (Figure 2.) and FC50-10 (Figure 5.) applied

Response 5:  The reviewer is correct, We have revised these errors, we confirm these samples as FAC10-50, so we revised the Figure 2 and Figure 5.

Point 6:  A careful proofread of the English, which is recommended, would improve the text, especially regarding to the use of the superscript and subscript (mm3, Ca(OH)2, Na+, P5+!). However, the reviewer suspects only some error in the transformation process of the submission.

Response 6:  The reviewer is correct, the superscript and subscript have been revised due to errors in the conversion process.

Reviewer 2 Report

This manuscript presents an investigation of the influence of a fly ash belite cement (FABC) addition on the hydraulic action of the P·I 42.5 Portland cement. This is a very relevant research subject from the perspective of environment protection taking into account the issues related to the utilization of low-quality fly ash that is the by-product of different manufacturing processes and grows ever more abundant. The authors performed comprehensive physicochemical investigations concerning the setting time, hydration temperature, mechanical strength after different exposition times, and the composition and microstructure of the hydration products formed in the Portland cement. They used the obtained data to determine what FABC addition to the P·I 42.5 cement is optimal and allows the strict standards for binding materials to be met. Their most important conclusion is that FABC effectively promotes the hydration of Portland cement and reduces the levels of early hydration heat release of silicate minerals as well as the total hydration heat. Both the writing and the overall quality of this experimental work are good. The title is an adequate reflection of the substance of the article, and the results will certainly be valuable to researchers working in the field. The manuscript can be recommended for publication after the authors address some minor concerns:

  1. Abstract: The authors write that "Notably, hydration of fly ash belite cement promotes formation of C-S-H gel, ettringite and calcium hydroxide, thereby significantly enhancing long-term strength." Which type of ettringite do the authors have in mind specifically? The "good" one or the "bad" one? The former is usually found in sulfate-slag and belite cements, which contain calcium sulfoaluminate. The latter forms in Portland and alumina cements. Although it is considered "bad", it is used to obtain expanding cements, so it is advantageous in some applications.
  2. Introduction, page 2, line 44: The last name of the lead author is different in the reference list (item [8]).
  3. Fig. 1: The labels used for the X-axis are different than what is used in the case of Fig. 2. I found the latter preferable. Please make these consistent.
  4. The impact of the porosity of the pastes and mortars on the mechanical strength of the cement should be discussed more thoroughly in the context of the available literature.
  5. When discussing the microstructure of the composite cement pastes, the authors mention that the products of hydration include C-S-H phases. The morphology type of this phase should be established much more precisely using the classification proposed by Diamond.

Author Response

Point 1:  Abstract: The authors write that "Notably, hydration of fly ash belite cement promotes formation of C-S-H gel, ettringite and calcium hydroxide, thereby significantly enhancing long-term strength." Which type of ettringite do the authors have in mind specifically? The "good" one or the "bad" one? The former is usually found in sulfate-slag and belite cements, which contain calcium sulfoaluminate. The latter forms in Portland and alumina cements. Although it is considered "bad", it is used to obtain expanding cements, so it is advantageous in some applications.

Response 1: Thanks for your attention, we have revised it as “ The reaction of Ca2+, SO42-, Al (OH)4-generating to AFt would consumes a lot of Ca (OH)2, which would destroyed the solid-liquid equilibrium of C3S and C2S. It promotes the hydration of pastes, releases more Ca2+, and finally improves the content of hydration products. Hence, The forming of ettringite is benefit to the hydration of composite cement.” in line 219-223.

Point 2: Introduction, page 2, line 44: The last name of the lead author is different in the reference list (item [8]).

Response 2: The reviewer is correct, we have revised it as “Shahsavari, R.; Chen, L.; Tao, L.; Edge dislocations in dicalcium silicates: Experimental observations and atomistic analysis. Cem. Concr. Res. 2016, 90, 80-88.”

Point 3: Fig. 1: The labels used for the X-axis are different than what is used in the case of Fig. 2. I found the latter preferable. Please make these consistent.

Response 3: The reviewer is correct,  we have revised as Fig. 3 and Fig. 4

Point 4: The impact of the porosity of the pastes and mortars on the mechanical strength of the cement should be discussed more thoroughly in the context of the available literature.

Response 4: Thanks for the suggestion,  we have revised as “ It has been indicated that [17] the high porosity was the main cause of low early strength of FABC pastes, and it was also not conducive to the development of early strength of composite cement. However, as the increase of FABC contents, the long-term strength would be improved with the densification of hydration products. In conclusion, the FABC content is is less than 30%, the long-term strength increase more obviously. ” in line 188-193.

Point 5: When discussing the microstructure of the composite cement pastes, the authors mention that the products of hydration include C-S-H phases. The morphology type of this phase should be established much more precisely using the classification proposed by Diamond.

Response 5:   Thanks for the suggestion,  we have revised as “ As Diamond proposed [23] that the morphology type of phases in cement have be established. There are more â…¡type of C-S-H gels that the network is connected into three-dimensional space by crossing the terminals, meanwhile, C3S and C2S were almost hardly visible on the surface of pastes. ” in line 240-244.

Round 2

Reviewer 1 Report

The manuscript has been improved remarkably, it can be accepted now.